# Water–Energy–Food Nexus Simulation: An Optimization Approach for Resource Security

**Albert Wicaksono, Gimoon Jeong and Doosun Kang \***

Department of Civil Engineering, Kyung Hee University, Yongin-si, Gyeonggi-do 17104, Korea;
albert.wcso@gmail.com (A.W.); gimoon1118@gmail.com (G.J.)
**\*** Correspondence: doosunkang@khu.ac.kr; Tel.: +82-031-201-2513

**Abstract:** The water–energy–food nexus (WEF nexus) concept is a novel approach to manage limited resources. Since 2011, a number of studies were conducted to develop computer simulation models quantifying the interlinkage among water, energy, and food sectors. Advancing a nationwide WEF nexus simulation model (WEFSiM) previously developed by the authors, this study proposes an optimization module (WEFSiM-opt) to assist stakeholders in making informed decisions concerning sustainable resource management. Both single- and multi-objective optimization modules were developed to maximize the user reliability index (URI) for water, energy, and food sectors by optimizing the priority index and water allocation decisions. In this study, the developed models were implemented in Korea to determine optimal resource allocation and management decisions under a plausible drought scenario. This study suggests that the optimization approach can advance WEF nexus simulation and provide better solutions for managing limited resources. It is anticipated that the proposed WEFSiM-opt can be utilized as a decision support tool for designing resource management plans.

**Keywords:** climate change; optimization; reliability index; resource management; WEF nexus

---

## 1. Introduction

Due to the rapidly increasing world population, water, energy, and food have become vulnerable resources. Global demand for these resources is increasing, while the available quantities are decreasing or limited. Currently, the world population is predicted to increase by up to 80 million per year and to reach 9.1 billion in 2050. This population growth will naturally increase the consumption of water, energy, and food [1]. The shift of air temperature due to climate change might increase the requirement of electricity for heating and cooling, while reducing the efficiency of electricity production, especially in thermal power plants [2]. On the other hand, climate change may cause severe droughts that limit water and food availability. Although hydraulic infrastructures, such as reservoirs (dams) and canals, may resolve the water shortage during the droughts, they require an efficient operation and proper allocation to accommodate the demand from various users.

Frameworks for integrated planning and management of resources, including Integrated Water Resources Management (IWRM) and Integrated Natural Resources Management (INRM), were implemented in policy-making processes to increase resource sustainability. Recently, the concept of the water–energy–food nexus (WEF nexus) was introduced as a new integrated approach that highlights the interdependence of these three sectors [3]. The concept was further expanded by involving other elements such as climate change, carbon emission, environment, and land availability [4,5]. Policy adaptation and involvement of multiple sectors (stakeholders, government, and private sectors) are also being raised as new topics in the WEF nexus discussion.

There are interconnections between the supply and demand of the water, energy, and food sectors. The water for energy interaction is the amount of water consumed in energy production. Examples of water for energy interactions include the cooling water used in thermal and nuclear power plants, the water used for hydropower, and the water required to grow bioenergy crops. On the other hand, examples of the energy for water interaction include the energy consumed for pumping, treating, and distributing water. Among water processes, seawater desalination is the most energy-intensive [3]. The water for food interaction is calculated based on the food's water footprint, which is the amount of water consumed to grow, produce, and process the food. Among food production processes, irrigation is the most water-intensive, accounting for 70% of global water consumption [6]. The application of pumps in an irrigation system is an example of the use of energy for food production. In addition, energy is consumed to produce fertilizers and pesticides and to operate agricultural machinery [7]. The use of food for energy production is described as bioenergy. Although bioenergy produces relatively few carbon emissions, it is one of the most water-intensive energy sources. The interconnections among WEF sectors can be conceptually described; however, the actual feedback connections are complex, often invisible, and affected by external factors.

To interpret and quantify the WEF nexus, several simulation models were developed and applied worldwide. Simulation models such as the WEF Nexus Tool 2.0 [8], MuSIASEM [9], and NexSym [10] were developed to quantify the nexus on national, regional, and local scales, respectively. Adopting concepts from the existing models, we developed a national-scale nexus simulation model, called the Water–Energy–Food Nexus Simulation Model (hereafter, WEFSiM) that quantifies and evaluates the interconnections between the water, energy, and food sectors [11]. The model was used to quantify resource consumption and production and, consequently, the future resource sustainability on a national scale. The model is useful for simulating plausible future scenarios and for creating suitable resource management plans. Based on the input parameters, the WEFSiM provides simulation results in terms of future resource availability and consumption. With the model, a user can simulate "what-if" scenarios and predict potential future conditions. In this study, we advance the WEFSiM by embedding an optimization module (WEFSiM-opt). WEFSiM-opt can support optimal decisions for stakeholders involved in creating sustainable resource management plans.

In this paper, a short review of the WEFSiM is firstly presented to give an overview of the conceptual framework, along with a general description of the steps used to create a simulation. A description of the optimization module (WEFSiM-opt) is provided in the following section. Finally, a simulation performed utilizing WEFSiM-opt is described, the results are presented, and conclusions are provided.

## 2. WEF Nexus Simulation Model (WEFSiM)

### 2.1. Conceptual Framework of WEFSiM

A detailed review of several existing nexus simulation models (WEF Nexus Tool 2.0, NexSym, and MuSIASEM) showed that the interconnections of WEF were not fully simulated in the existing models [11,12]. The WEF Nexus Tool 2.0 simulates the requirements of water, energy, and farmland to support self-sufficiency of food products but provides limited feedback analysis between resources [8]. The NexSym model provides a more comprehensive analysis but is only capable of local-scale simulation [10]. Meanwhile, the MuSIASEM simulates the WEF nexus with consideration of external components such as land, economy, human capital, ecosystem, greenhouse gas emissions, and land use based on Georgescu-Roegen's flow-fund element approach [9]. Nevertheless, none of the existing models are equipped with an optimization module and, thus, require the user to perform multiple manual simulations to find the best overall management plan.

In the WEFSiM, feedback analysis among the resources is realized based on the concepts of "actual availability" and "indirect demand". For example, the actual availability of treated water depends on the indirect demand of energy for water (E4W), while the water for energy (W4E) is determined based

on the required energy production. The feedback connections also affect other resources in various ways, which can be summarized by, but are not limited to, the conceptual diagram shown in Figure 1. The representative feedback connections can be described as follows:

- The raw water supply includes surface water, groundwater, wastewater, and seawater. An example of a direct demand for raw water is the need to send raw water to treatment plants so that it can be used by municipal and industrial users. Examples of indirect demand include water for irrigation for food production (W4F) and cooling water for energy-generation processes (W4E).
- Electric energy is generated from four types of power plants: hydro, nuclear, thermal, and renewable energy. There is a direct demand for energy from municipal and industrial users. Some examples of indirect demand are energy for water (E4W) for things such as water treatment and pumping operations, and energy for food (E4F) for irrigation, production of fertilizer and agrichemicals, etc.
- Food is categorized as meat-based (e.g., bovine meat, poultry meat, and eggs) or crop-based (e.g., rice, vegetables, and fruits). The available supply of food meets the direct demand in the form of food for eating and the indirect demand in the form of bioenergy, which is food for energy (F4E).

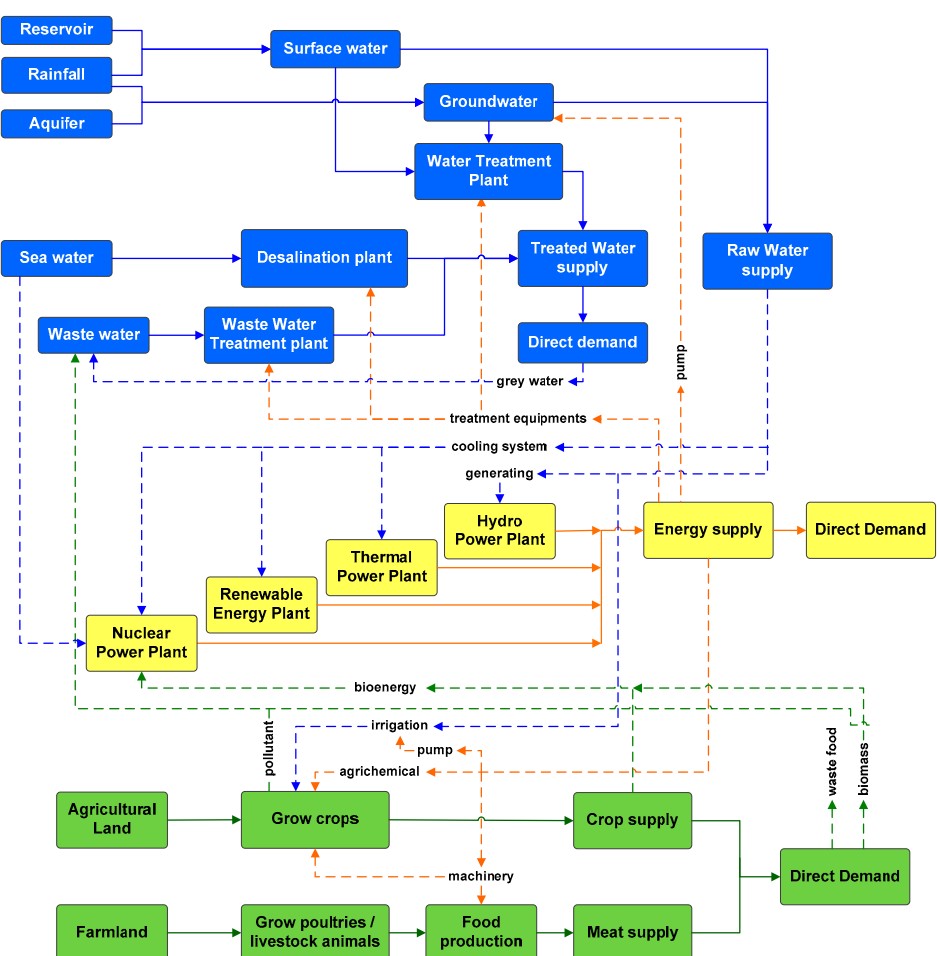

**Figure 1.** The conceptual framework of the water–energy–food (WEF) nexus simulation model (WEFSiM).

### 2.2. WEFSiM Simulation Steps

The WEFSiM consists of five main calculation steps as illustrated in Figure 2.

- Step 1 (Initialization): WEFSiM imports information about the study area from the database. The spreadsheet-type database consists of statistical data including production and consumption

of resources and resource production intensity. The resource production intensity is a conversion factor that determines the required resource in the production of a unit of other resources. Future conditions can be included in the input and can be extrapolated based on the historical information available for the study area. Several input parameters (e.g., agricultural area, population growth, resource consumption and production rates, and resource allocation priorities) can be altered to perform varying scenario-based simulations.

- Step 2 (Demand calculation): Total demand is calculated as the sum of the direct and indirect demand. The direct demand is the amount consumed by the domestic population and industry, and it is calculated as the product of consumption rate and population or gross domestic product (GDP). Meanwhile, the indirect demand represents the consumption of a resource in the production of other resources, and it is calculated based on production amount and resource production intensity.

- Step 3 (Potential availability calculation): Potential availability is defined as the maximum resource production capacity for the study area. In the water sector, the potential available water (PAW) is determined by the rainfall and stored water in the reservoir. The potential available treated water (PATW) is determined based on the capacity of the water treatment plant. Then, the potential available raw water (PARW) is determined by the difference between the potential available water (PAW) and the PATW as expressed in Equation (1). For energy, the potential available energy (PAE) is defined as the maximum capacity of the power plant. In the food sector, the potential available food (PAF) is calculated based on the agricultural land area (AgL) and its production rate (prf) as expressed in Equation (2).

$$PARW = PAW - PATW. \tag{1}$$

$$PAF = AgL \times prf. \tag{2}$$

- Step 4 (Resource allocation): Resource allocation begins by sorting the consumers based on predefined supply priority. Then, the potential available resources are allocated; higher-priority consumers are supplied first, with lower-priority consumers following sequentially. The available resources may not always be enough to supply the consumer demand; thus, the actual supply amount is determined as the minimum between available supply and required demand. Since the actual production can be different from the potential availability, indirect demand and actual demand should be recalculated, which requires a reallocation of resource supply. Therefore, several core equations are modified. For the water sector, the actual available treated water (AATW) is determined by the PATW and energy-dependent available treated water (ATW(e)), as expressed in Equation (3). Since the energy supply for water might be limited, the ATW(e) can be smaller than the PATW. A similar relationship applies to energy and food sectors, as seen in Equations (4) and (5). Actual available energy (AAE) is calculated as the minimum of PAE and water-dependent available energy (AE(w)), as shown in Equation (4). In the food sector, the actual available food (AAF) is determined by agricultural land area (PAF), energy-dependent food production (AF(e)), water-dependent food production (AF(w)), and reduction factor (r), as expressed in Equations (5) and (6). The reduction factor (r) is defined in a range of 0 to 1 to represent the crop tolerance to drought. Note that each crop or livestock has a different level of tolerance. A high reduction factor represents low tolerance to drought. In this step, system dynamics analysis is implemented to calculate the actual demand and supply simultaneously.

$$AATW = min(PATW, ATW(e)). \tag{3}$$

$$AAE = min(PAE, AE(w)). \tag{4}$$

$$AF = min(PAF, AF(e), AF(w)). \tag{5}$$

$$AAF = AF + (1 - r) \times (PAF - AF).　\tag{6}$$

- Step 5 (Reliability calculation): In this step, the overall shortage or excess of resources is determined, and the user reliability index (URI) is calculated based on the actual demand and supply determined in Step 4. The URI is calculated as a ratio of actual supply and demand for each resource over the entire simulation period, as expressed in Equation (7). The URI is calculated individually for each resource sector (i.e., water, energy, and food), and total reliability index ($RI_{tot}$) across multiple resource sectors is calculated, as expressed in Equation (8). The index value ranges from 0 to 1, and a reliability index of 1 indicates that the relevant resource is fully supplied without shortage.

$$\text{User Reliability Index (URI)} = \frac{1}{T} \sum_{t=1}^{T} \left( \frac{\text{Actual Supply (t)}}{\text{Actual Demand (t)}} \right), \tag{7}$$

$$\text{Total Reliability Index (RI}_{tot}) = \frac{1}{N} \sum_{n=1}^{N} \text{URI}(n), \tag{8}$$

where T = simulation period, and N = number of resources sectors.

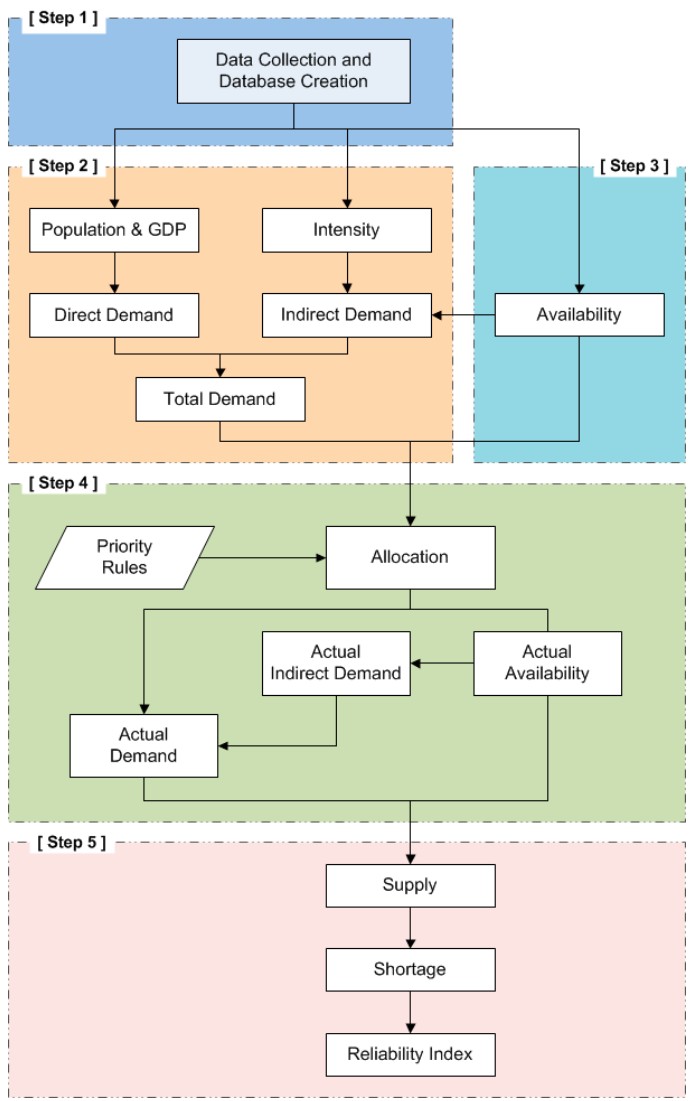

**Figure 2.** WEFSiM simulation steps.

More details about the development and implementation of WEFSiM can be found in a technical paper by the authors [11]. The paper also presents the development of the database and the model calibration process. The model calibration was conducted using the historical data of water withdrawal, energy production and consumption, and food production and consumption. Overall, the error between the reported data and model results was less than 6.0%, indicating acceptable model performance [11]. The application of WEFSiM resulted in a successful simulation of resource supply, resource demand, and reliability indices in two study areas (Korea and Indonesia) that differ in resource availability and economic environment. Several plausible future scenarios were developed and simulated as a case study to predict future resource conditions, given scenarios for future natural variations (e.g., climate changes) and societal variations (e.g., population fluctuations).

## 3. Optimization Module for WEFSiM (WEFSiM-Opt)

### 3.1. Optimization Algorithm

Optimization is a technique to find optimal solutions by adjusting decision variables to minimize or maximize the objective function(s) [13]. A conventional approach to finding the optimal solution of the objective function is to use trial-and-error, based on manually altering the decision values. In practice, the optimization process involves many decision variables and complicated calculation steps. Thus, a computer model is helpful in performing the steps, as long as the variables and functions can be adequately expressed in computer code.

For decades, a number of optimization approaches were developed to solve engineering problems. Among evolutionary search algorithms, the genetic algorithm (GA) is widely utilized in research and practice. The GA adopts evolutionary theory in which the stronger species survive, while the weaker ones are eliminated by natural selection, which creates stronger offspring in succeeding generations [14]. In general, GA begins with determining the initial parents either by random selection or using a pre-defined initial population. Then, the parents are scored to determine which are best to generate offspring. The offspring are generated by one of the following methods: elitism (based on the best score), crossover (combining two parents), and mutation (randomly mutating the parent). The elitism method ensures that the good offspring survive through the generations, the crossover method leads a population to converge to the optimal solution, and the mutation method introduces diversity to avoid being caught in a local optimum.

In the early stages of development, GA was only able to find an optimal value for a single fitness function. This type of GA is known as a single-objective genetic algorithm (SOGA). Later, Schaffer [15] introduced a new approach that found optimal solutions for multi-objective problems using a vector-evaluated genetic algorithm (VEGA); however, this method was only capable of finding the extreme solutions of each objective function. Since then, a series of multi-objective optimization algorithms were developed, including a multi-objective genetic algorithm (MOGA).

The primary purpose of the multi-objective optimization is to find Pareto optimal solutions that satisfy multiple objective functions simultaneously [16]. The Pareto is a set of optimal solutions none of which is dominant over the others [14]. The MOGA implements a procedure similar to GA, starting with a set of random or pre-defined initial populations. The same processes (i.e., elitism, crossover, and mutation) are applied. However, with MOGA, the next generation is evaluated based on the "non-dominated" rank and distance measure for such individuals in the current generation. The Pareto fraction and distance function are two additional parameters of MOGA that maintain individuals with higher fitness and the population diversity to achieve Pareto optimal solutions. The pareto fraction limits the number of elite members in each generation, while the distance function helps maintain the diversity on a Pareto front by keeping the outlying individuals [17]. The MOGA returns all individuals in the Pareto front and, thus, can provide multiple solutions (alternatives). This is helpful, but users may require further interpretation to choose a preferred solution. The WEFSiM-opt is equipped with

SOGA and MOGA, and the optimization results of the two algorithms are analyzed and compared in the application study described below.

*3.2. Objective Functions and Decision Variables*

The objective function of WEFSiM-opt is to maximize the URI of water, energy, and food and the total reliability index ($RI_{tot}$) as presented in Equations (7) and (8), respectively. The proposed reliability index quantifies the availability of resources to supply the demand, which is directly affected by the decision variables of the optimization models. The index ranges from 0 to 1; thus, it is convenient as an objective function and easy to interpret the results. The four objective functions (i.e., the three URIs and $RI_{tot}$) can be optimized individually in the SOGA or simultaneously in the MOGA.

The decision variables comprise 21 parameters, including the priority index (eight decisions) and the water allocation portions (13 decisions), which are explained below. Optimal decisions of the selected variables maximize the resources reliability under limited resources.

- The priority rules consist of eight variables that determine the importance of water and energy users. A priority index of 1 is the highest and will be supplied first, followed by the lower-priority users, in a sequential manner. If there are two or more users with the same priority, the available resources will be distributed proportionally based on their requirements. This study categorizes water users into four groups of municipal, agricultural, energy, and industrial sectors. Energy users are categorized into municipal, water, agricultural, and industrial groups.
- The decisions within the water allocation sector determine the water source from which individual users are supplied. Surface and groundwater are supplied to all four users (i.e., municipal, agriculture, energy, and industry), while reclaimed water can only be supplied to three users (i.e., municipal, agriculture, and industry). Desalinated water can be supplied for municipal and industrial purposes, while raw seawater is only supplied as cooling water in the energy sector.

*3.3. Optimization Module Development*

The optimization module is linked to the simulation module without changing the overall calculation steps, as illustrated in Figure 3. The optimization starts with initialization based on the database, the predefined priority index, and water allocation. Then, it follows the simulation steps to calculate the URIs and $RI_{tot}$ of the initial data. If the WEFSiM-opt finds a possibility to improve the results, it will alter the decision variables (priority index and water allocation) automatically and recalculate the objective functions (URIs and $RI_{tot}$). The optimization module iterates the calculation process by modifying the decision variables (via elitism, crossover, and mutation) until the maximum URIs and $RI_{tot}$ are obtained or a pre-defined maximum generation is reached. The WEFSiM-opt can perform both the SOGA and the MOGA. In the SOGA, optimization of the four objective functions (i.e., the three URIs and $RI_{tot}$) is executed separately, while the MOGA maximizes the four objectives in a single optimization run.

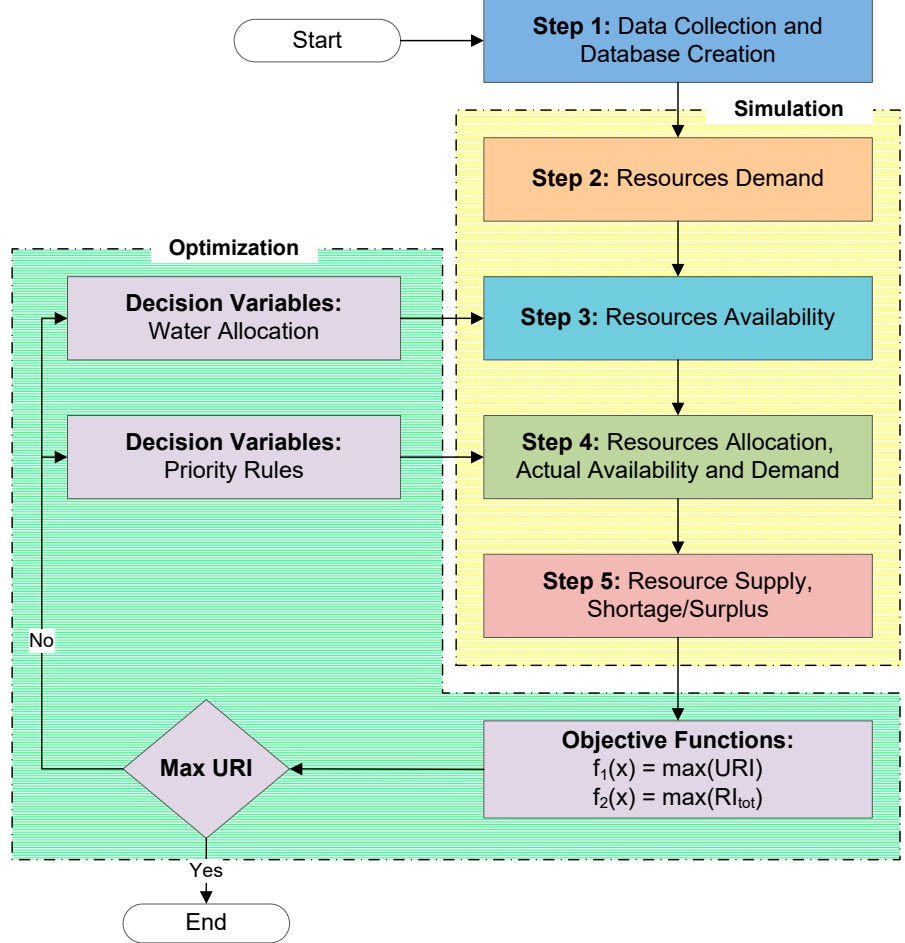

**Figure 3.** Optimization steps of WEFSiM-opt.

## 4. Model Applications

### 4.1. Scenario Development

A scenario was designed to represent plausible future conditions in Korea using the WEFSiM-opt. The plausible future conditions were predicted based on historical information collected from the Food and Agriculture Organization of United Nations (FAO) database for food and agriculture (FAOSTAT), FAO's information system of water and agriculture (AQUASTAT), the World Bank, and other research institutes listed in the technical paper by Wicaksono and Kang [11]. The collected historical data from 2000 to 2012 were extended 25 years into the future (through 2037), as shown in Figure 4. Three regression formulas (linear, polynomial, and exponential equations) were applied for extrapolation, and the best-fit curve was selected for each dataset. As seen in Figure 4, the linear model showed the most logical regression results in most cases, while the polynomial equation was the best fit for the production rate of several food types. Please note that the determination of the best-fit curve is not only based on the coefficient of determination ($R^2$) but also considering the reasonable projected values. The annual rainfall was forecast under the climate scenario of RCP4.5, which depicts the reduction of greenhouse gas emissions due to the declination of fossil-fuel usage and increment of renewable energy consumption [18]. Among the datasets generated by several institutions, including Norwegian Climate Centre (NCC, Norway), Beijing Climate Center (BCC, China), Centre National de Recherches Météorologiques - Centre Européen de Recherche et de Formation Avancée en Calcul Scientifique (CNRM-CERFACS, France), The Euro-Mediterranean Center on Climate Change (CMCC, Italy), and Canadian Centre for Climate Modelling and Analysis (CCCma, Canada), the lowest

annual rainfall prediction (BCC) was selected to simulate the potential drought scenario, as presented in Figure 4D.

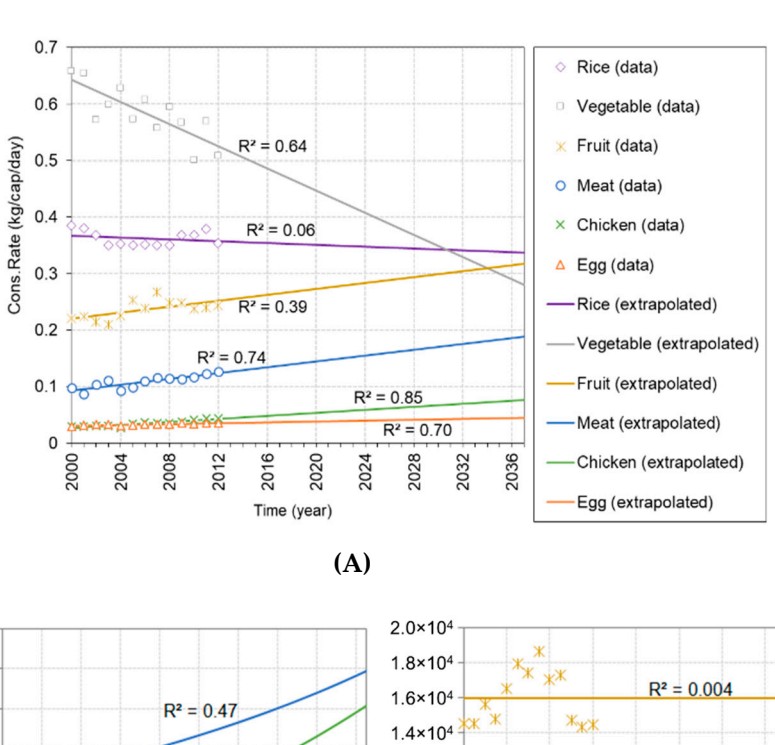

**(A)**

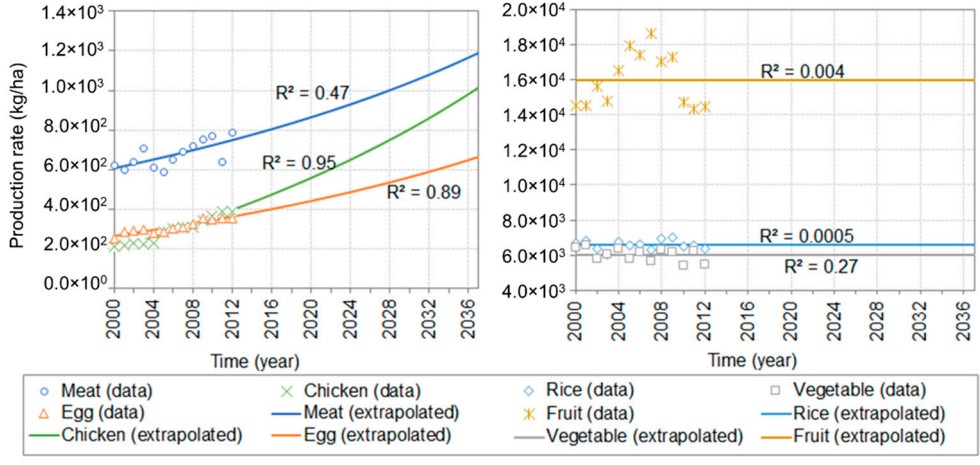

**(B)**

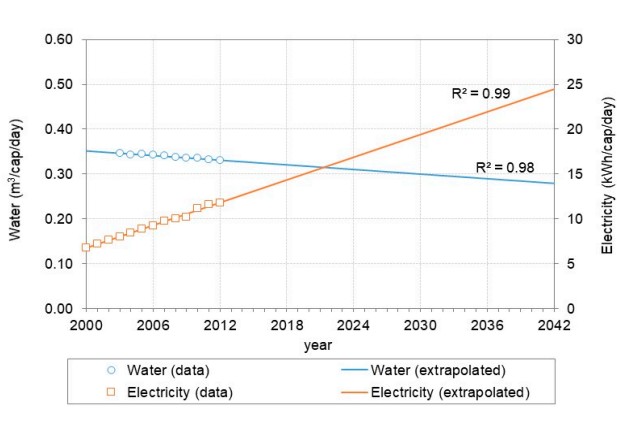

**(C)**

**Figure 4.** *Cont.*

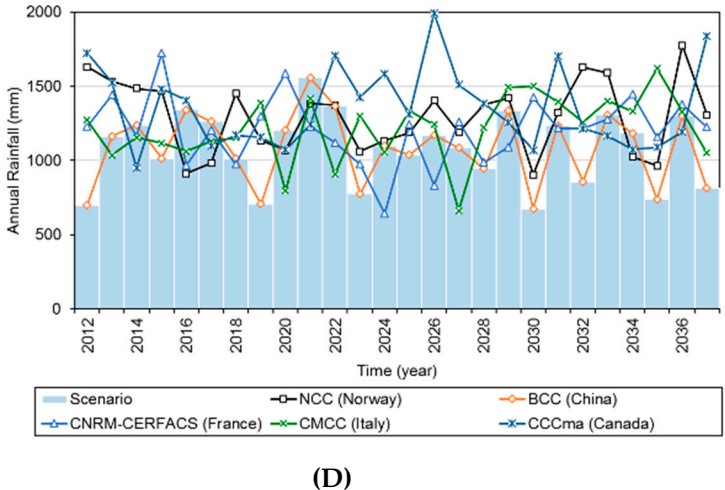

**(D)**

**Figure 4.** Prediction of (**A**) food consumption rate, (**B**) food production rate, (**C**) water and energy consumption rate, and (**D**) annual rainfall for the study area (Korea).

*4.2. Scenario Results*

4.2.1. Comparison of SOGA and MOGA Solutions

Both the SOGA and MOGA were performed to maximize the URI of water (URI_W), energy (URI_E), and food (URI_F) and the total reliability index ($RI_{tot}$) by determining the optimal priority index and water allocation portions. In SOGA, the four objective functions were maximized individually to obtain a single optimal solution. The MOGA maximized the four objective functions simultaneously in a single optimization run to produce a set of Pareto optimal solutions. The maximum URIs obtained from the two optimizations are depicted in Figure 5, which illustrates the URI pairs of water–energy, water–food, and energy–food. Meanwhile, Table 1 summarizes the optimal decision variables corresponding to the maximum URIs of water, energy, and food and the $RI_{tot}$ found by the SOGA and MOGA. Note that the presented MOGA solutions were selected among the Pareto solutions. Unlabeled dots depict the Pareto solutions in Figure 5.

Analyzing the results, the SOGA and MOGA both placed the food sector at the lowest priority order when allocating water and energy to achieve maximum URIs of water and energy except when maximizing URI_F. It is interesting to note that the decisions to maximize URI_E also maximized $RI_{tot}$. As seen in Table 1, the SOGA and MOGA provided almost identical decisions for individual objectives; however, the MOGA could balance the results by adjusting the water allocation. For example, when optimizing the food supply (i.e., maximizing URI_F), MOGA increased the water supply for the energy sector to increase energy production, which consequently affected the treated water production (Figure 6) and improved URI_W compared to the SOGA solutions maximizing URI_F. Overall, the MOGA is superior to the SOGA since it provides solutions with greater balance and without bias to a certain resource.

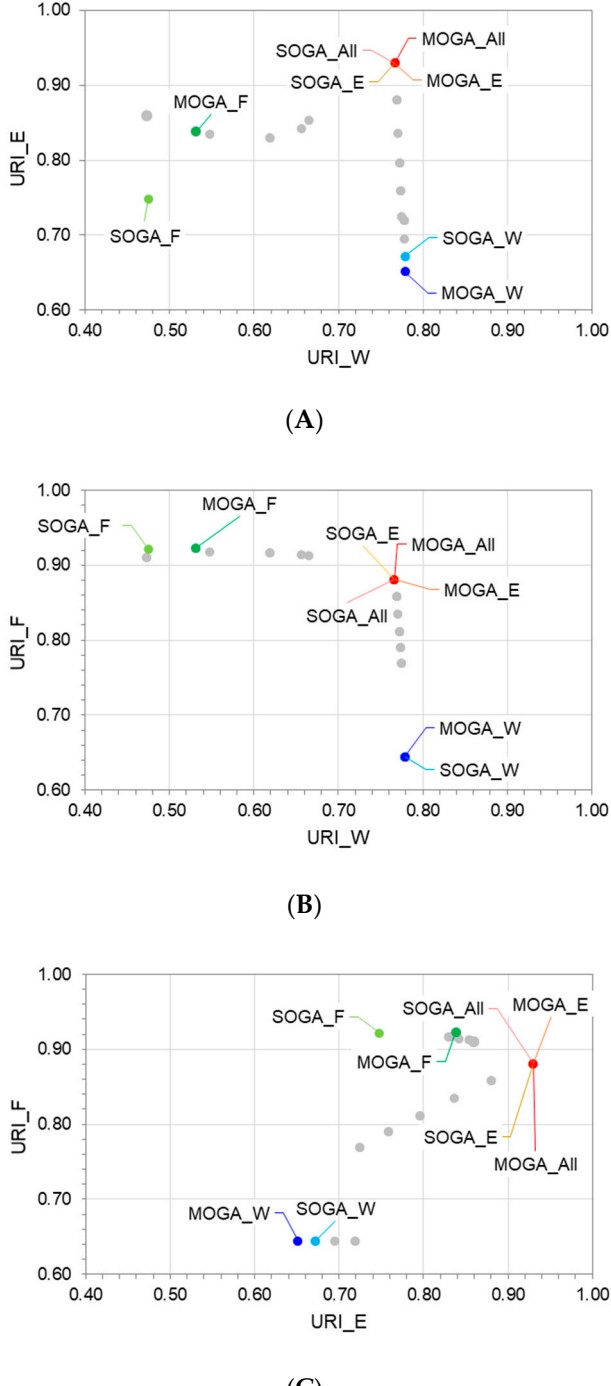

**Figure 5.** Maximum user reliability indices (URIs) obtained from single-objective genetic algorithm (SOGA) and multi-objective genetic algorithm (MOGA) for (**A**) water–energy, (**B**) water–food, and (**C**) energy–food pairs.

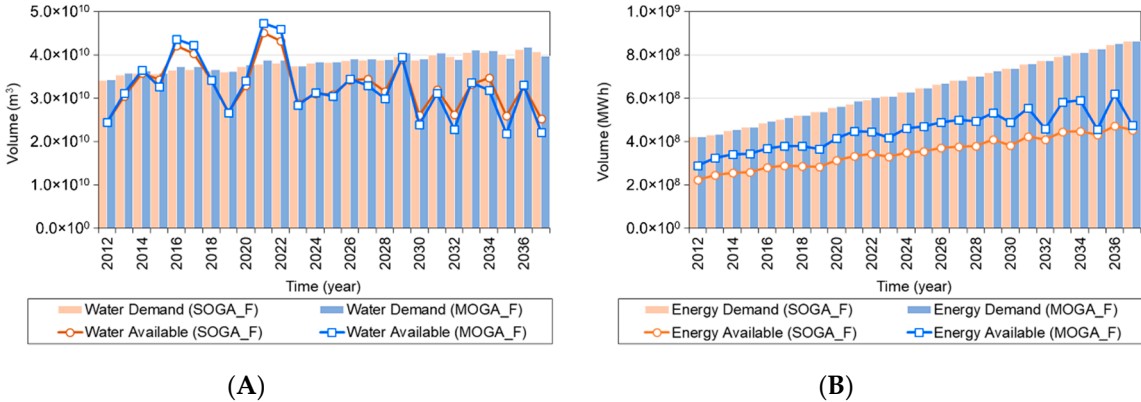

**Figure 6.** Comparison of SOGA and MOGA solutions maximizing URI_F for availability and demand of (**A**) water and (**B**) energy sectors.

**Table 1.** Optimal solutions from single-objective genetic algorithm (SOGA) and multi-objective genetic algorithm (MOGA). W—water; E—energy; F—food; URI—user reliability index.

| Parameter | | SOGA | | | | MOGA | | | | |
|---|---|---|---|---|---|---|---|---|---|---|
| | | W | E | F | Total | W (G3) | E (G2) | F (G1) | Total (G2) | G4 |
| **Priority Index** | | | | | | | | | | |
| Water | Municipal | 1 | 1 | 1 | 1 | 1 | 1 | 1 | 1 | 1 |
| | Food | 3 | 2 | 1 | 2 | 3 | 2 | 1 | 2 | 2 |
| | Energy | 2 | 1 | 1 | 1 | 2 | 1 | 1 | 1 | 1 |
| | Industry | 1 | 1 | 2 | 1 | 1 | 1 | 2 | 1 | 1 |
| Energy | Municipal | 1 | 1 | 1 | 1 | 1 | 1 | 1 | 1 | 1 |
| | Water | 1 | 1 | 1 | 1 | 1 | 1 | 1 | 1 | 1 |
| | Food | 3 | 2 | 1 | 2 | 3 | 2 | 1 | 2 | 2 |
| | Industry | 2 | 2 | 2 | 2 | 2 | 2 | 2 | 2 | 2 |
| **Water Allocation (%)** | | | | | | | | | | |
| Surface water | Municipal | 69.0 | 69.0 | 69.0 | 69.0 | 69.0 | 69.0 | 69.0 | 69.0 | 69.0 |
| | Food | 80.0 | 80.0 | 80.0 | 80.0 | 80.0 | 80.0 | 80.0 | 80.0 | 80.0 |
| | Energy | 69.3 | 70.0 | 60.0 | 70.0 | 68.4 | 70.0 | 66.6 | 70.0 | 68.1 |
| | Industry | 55.0 | 55.0 | 55.0 | 55.0 | 55.0 | 55.0 | 55.0 | 55.0 | 55.0 |
| Ground water | Municipal | 10.0 | 10.0 | 10.0 | 10.0 | 10.0 | 10.0 | 10.0 | 10.0 | 10.0 |
| | Food | 15.0 | 15.0 | 15.0 | 15.0 | 15.0 | 15.0 | 15.0 | 15.0 | 15.0 |
| | Energy | 8.6 | 10.0 | 7.0 | 10.0 | 9.3 | 10.0 | 9.0 | 10.0 | 8.6 |
| | Industry | 15.0 | 15.0 | 15.0 | 15.0 | 15.0 | 15.0 | 15.0 | 15.0 | 15.0 |
| Reclaim water | Municipal | 20.0 | 20.0 | 20.0 | 20.0 | 20.0 | 20.0 | 20.0 | 20.0 | 20.0 |
| | Food | 5.0 | 5.0 | 5.0 | 5.0 | 5.0 | 5.0 | 5.0 | 5.0 | 5.0 |
| | Energy | 0.0 | 0.0 | 0.0 | 0.0 | 0.0 | 0.0 | 0.0 | 0.0 | 0.0 |
| | Industry | 30.0 | 30.0 | 30.0 | 30.0 | 30.0 | 30.0 | 30.0 | 30.0 | 30.0 |
| Desalinated water | Municipal | 1.0 | 1.0 | 1.0 | 1.0 | 1.0 | 1.0 | 1.0 | 1.0 | 1.0 |
| | Food | 0.0 | 0.0 | 0.0 | 0.0 | 0.0 | 0.0 | 0.0 | 0.0 | 0.0 |
| | Energy | 0.0 | 0.0 | 0.0 | 0.0 | 0.0 | 0.0 | 0.0 | 0.0 | 0.0 |
| Sea water | Energy | 20.0 | 20.0 | 20.0 | 20.0 | 20.0 | 20.0 | 20.0 | 20.0 | 20.0 |
| **Reliability Index** | | | | | | | | | | |
| URI_Water | | 0.778 | 0.766 | 0.475 | 0.766 | 0.779 | 0.766 | 0.532 | 0.766 | 0.772 |
| URI_Energy | | 0.672 | 0.930 | 0.748 | 0.930 | 0.651 | 0.930 | 0.838 | 0.930 | 0.796 |
| URI_Food | | 0.645 | 0.881 | 0.922 | 0.881 | 0.645 | 0.881 | 0.923 | 0.881 | 0.812 |
| RI_Total | | 0.698 | 0.859 | 0.715 | 0.859 | 0.692 | 0.859 | 0.764 | 0.859 | 0.793 |

### 4.2.2. Grouping MOGA Solutions

Detailed observation showed that the MOGA produced 55 Pareto solutions with different reliability indices. We implemented the self-organizing map (SOM) and classified the 55 solutions into four groups based on emphasizing reliability index, as shown in Figure 7. For further analysis,

a representative solution was selected from each group. The representative solutions of groups 1, 2, and 3 were the solutions maximizing the URIs of food, energy, and water, respectively. The representative solution of group 4 was the one closest to the optimal point (point [1,1,1] in Figure 7).

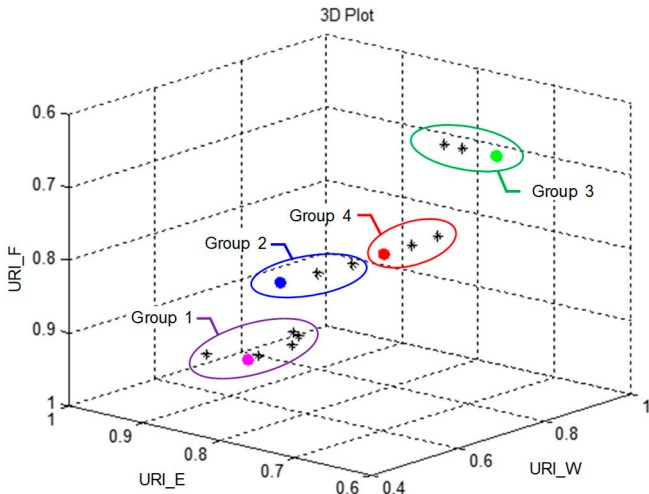

**Figure 7.** MOGA Pareto solutions and four groups based on self-organizing map (SOM) clustering analysis.

As briefly described in the previous section, varying the combinations of priority index and water allocation affected the availability and demand of resources, which consequently affected the reliability index, as shown in Figure 8D. The paragraphs below describe the findings and characteristics of each MOGA solution group.

- General results Rainfall forecasting based on the RCP4.5 scenario showed low annual rainfall in the future and possible droughts. This was indicated by lower availability of water than demand, as shown in Figure 8A. Also, the energy supply was predicted to decrease in drought years (Figure 8B) due to a lack of water supply for energy production. The cooling water supply from seawater was considered only for seashore power plants that utilize seawater. The drought condition also affected the food sector, since food production is highly dependent on water availability, as shown in Figure 8C. Moreover, agriculture was the most water-consuming sector, averaging 68% of total water use. Please note that the notations of G1, G2, G3, and G4 in Figure 8 are designations for the four solution groups.

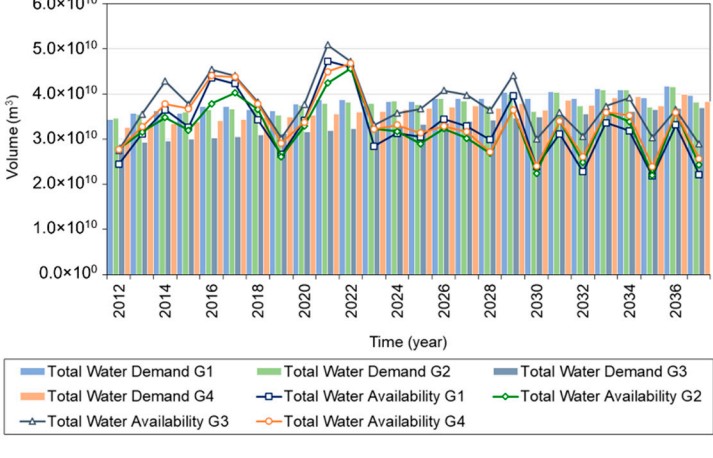

**(A)**

**Figure 8.** *Cont.*

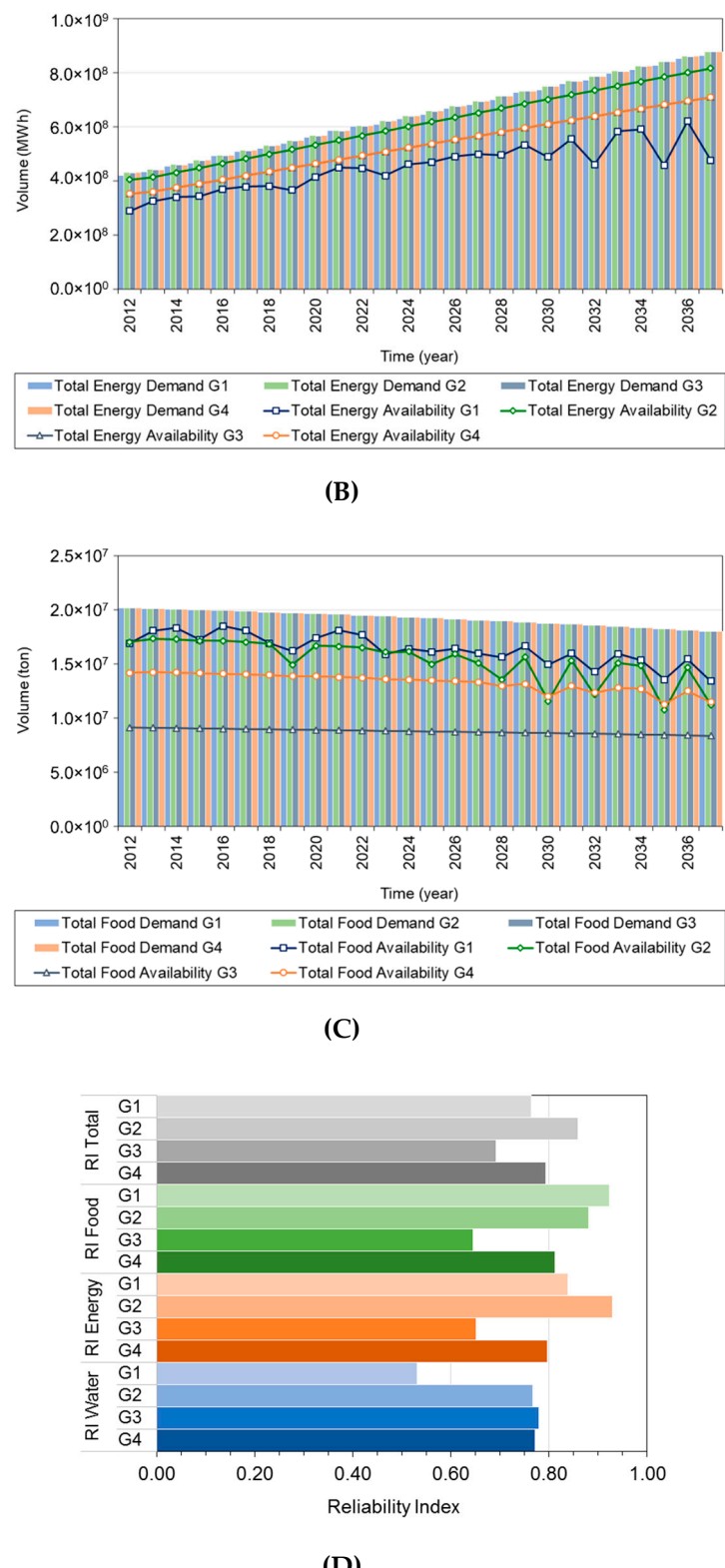

**Figure 8.** Representative solutions of the four groups showing results of (**A**) water, (**B**) energy, (**C**) food demand and supply, and (**D**) URIs.

- Group 1 (G1) The solutions of Group 1 were designed to maximize URI of food by placing the food sector at the highest priority level when allocating water and energy. The solution

consequently affected the water and energy sectors, while URI_W in Group 1 was significantly reduced compared with the other groups, as seen in Figure 8D.

- Group 2 (G2) The solutions of Group 2 were designed to maximize the URI of energy, which also maximized the total RI. The solutions assigned the lowest priority to the most consumptive consumers (i.e., industry for energy supply, and food for water and energy supply). Thus, more water and energy were supplied for other consumers, which increased the reliability of the supply overall.

- Group 3 (G3) The solutions within Group 3 were designed to maximize the URI of water by placing the food sector at the lowest priority level for water and energy supply. The food sector was the largest consumer of water. The solutions were designed to conserve water to increase water reliability but resulted in the lowest reliability of all resources, as well as the lowest $RI_{tot}$, as seen in Figure 8D.

- Group 4 (G4) In contrast to the other groups, the solutions in Group 4 were designed to balance the supply of all resources, rather than emphasizing a specific resource. The priority index set was the same as for Group 2; however, the allocation of water for energy was reduced to conserve water. The results showed decreases in energy and food production and a slight increase in water reliability.

A user of WEFSiM-opt may explore other Pareto solutions in addition to the representative solutions listed in Table 1. Analysis of the various alternatives in the Pareto group reveals advantages and disadvantages for each solution and can help decision-makers establish an appropriate resource management plan.

4.2.3. Understanding the Interconnections between Resources Using WEFSiM-Opt

The interconnections between resources can be described with either a tradeoff or a feedback connection. Tradeoff occurs when two resources are compromising for each other, that is, the production of a particular resource will sacrifice another resource, or vice versa. The feedback connection is a relationship that occurs between resources that sequentially influence each other (e.g., availability of treated water that is determined by the energy availability). Here, the resource tradeoff was evaluated based on the URI_W and URI_F of Group 1 and Group 3. As listed in Table 1, the solution in Group 1 (G1) resulted in the maximum URI_F and the minimum URI_W, while the solution in Group 3 (G3) produced the opposite result. The solutions of G1 tended to supply more water for agriculture than for energy, which reduced the water supply to the power plants and limited the energy supply for water treatment (Figure 9A,B). Thus, the URI_W was reduced because the treated water could not fulfill the demand. Meanwhile, since the solutions in G3 emphasized the URI_W, more water was allocated for energy production, which increased both the energy supply for water treatment and the availability of treated water (Figure 9C). However, it reduced the water supply for agriculture and lowered the URI_F. Based on this example, it is clear that the production of treated water and food is dependent on the water allocation decisions for the energy and food sectors. Due to a limited supply of raw water in drought years, the water, energy, and food sectors could not be fully supplied at the same time. By understanding and quantifying the tradeoff between resources, stakeholders can analyze the various risks that may occur under adverse conditions.

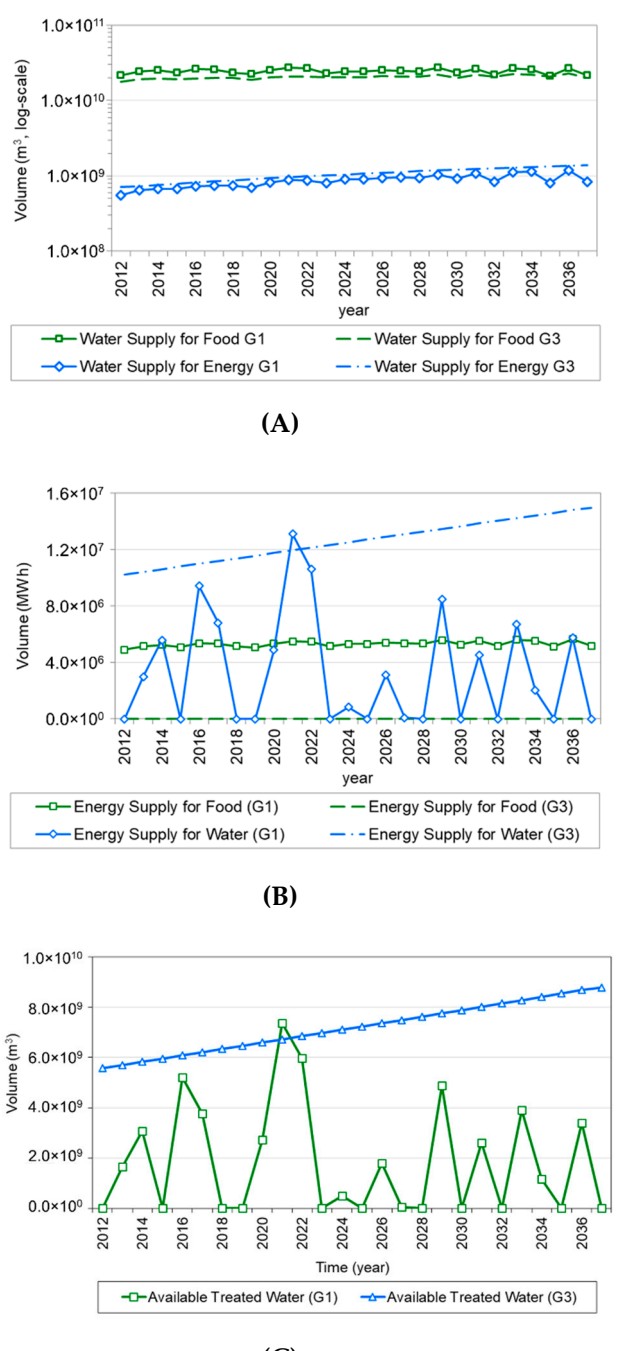

**Figure 9.** Comparison of (**A**) water supply, (**B**) energy supply, and (**C**) available treated water between Group 1 and Group 3.

Meanwhile, the feedback connection between resources was observed by evaluating the food availability of a solution in Group 2 (G2), as illustrated in Figure 10. Note that food production was limited by availability of farmland area, water, and energy (Equation (5)). As seen in Figure 10, food production was mainly limited by available energy (AF(e)), while that during drought years (e.g., 2029–2037) was affected by water availability (AF(w)). Feedback analysis is useful to identify the critical elements in producing specific resource changes. Knowledge of both the tradeoff and feedback between resources is critical for preparing adequate management plans under various scenarios.

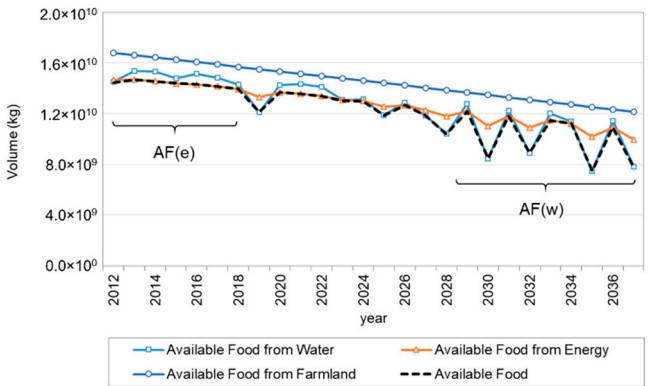

**Figure 10.** Critical factors affecting food production in Group 2.

*4.3. Discussion*

The WEF nexus simulation models, such as WEF Nexus Tool 2.0, MuSIASEM, NexSym, and WEFSiM, were implemented to simulate and evaluate possible scenarios and to propose solutions to establish a sustainable resources management. According to the RCP4.5 scenarios, potential droughts are anticipated in the future. Several water management scenarios might be conducted to cope with this issue. However, current simulation models require multiple simulations to evaluate different management scenarios to find suitable solutions. Indeed, it is a quite intensive process and does not guarantee optimal decisions. Therefore, the optimization module was introduced and embedded in WEFSiM to maximize the reliability index (URIs and $RI_{tot}$) by optimizing the management decisions. Two optimization algorithms (SOGA and MOGA) successfully provided sets of solutions to maximize the reliability of each resource. Based on the solution sets, stakeholders could designate the priority index and water allocation amount to manage the limited resources. For example, using the priority index, stakeholders can prioritize the user that produces higher benefit. From the application study, the results suggest that the stakeholders can choose the priority index of Group 1 to increase food production, or they can choose the solution of Group 3 during the drought seasons since it maximizes the water availability. Meanwhile, the water allocation decision is more related to the conservation of water in the area. It may help stakeholders to optimize the available water usage to supply all demands. The four solution groups intend to ease the burden on stakeholders when determining the management plans under a certain circumstance. However, the selection of a management plan that emphasizes a certain resource might influence other resources. Therefore, a closer look for tradeoff and feedback analysis between resources should be implemented to identify and anticipate the possible impacts.

Note that both WEFSiM and WEFSiM-opt were developed for a nation-scale simulation. In reality, a nation-scale area consists of multiple local regions/watersheds that have different resource characteristics. Future enhancement of WEFSiM-opt will include the regional resource evaluation and resource transfer among regions, which should consider an economic analysis for capital investment decisions.

**5. Conclusions**

The WEF nexus optimization model (WEFSiM-opt) was developed to advance the previously developed nexus simulation model (WEFSiM). The WEFSiM-opt is equipped with an optimization module that consists of single- and multi-objective optimization algorithms to provide optimal decisions for water allocation and the priority index maximizing resource supply. The proposed model was applied to a case study of Korea under a drought scenario, due to the potential for climate change in the future. In this study, both the single-objective and multi-objective genetic algorithms (SOGA and MOGA, respectively) were implemented to maximize four URIs. The MOGA provided more balanced solutions than the SOGA. However, the MOGA required further analysis to evaluate the obtained Pareto solutions, which were clustered into four groups with different characteristics

and URI results. The solutions from the individual groups might ease the burden on stakeholders when choosing the proper decisions that are appropriate with the characteristics of the planning area. Based on these groups, further analysis was also conducted to identify the tradeoff and feedback connections among resources. This analysis revealed a detailed correlation among resources, which could predict and evaluate the effect from implementation of a certain alternative. An understanding of these correlations would be useful for stakeholders in determining appropriate resource management plans under diverse future scenarios.

This study suggests that the optimization approach for WEF nexus simulations can increase the accuracy of models created for managing limited resources. To our knowledge, this is the first attempt to apply an optimization scheme in a WEF nexus modeling study. The proposed WEFSiM-opt can be utilized as a decision support tool for engineers and decision-makers involved in designing resource management plans. Furthermore, the additional modules to simulate resources transfer among local areas with economic analysis should be added in the model for enhanced simulations.

**Author Contributions:** Conceptualization, D.K.; data curation, G.J.; formal analysis, A.W.; methodology, A.W.; software, A.W. and G.J.; supervision, D.K.; writing—original draft, A.W.; writing—review and editing, D.K.

**Funding:** This study was supported by the Korea Ministry of Environment with the project "Graduate School specialized in Climate Change" and by the Korea Environmental Industry & Technology Institute (KEITI) grant funded by the Ministry of Environment (Grant RE201901084).

**Conflicts of Interest:** The authors declare no conflict of interest.

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
