# Peer review of "Water–Energy–Food Nexus Simulation: An Optimization Approach for Resource Security"

_water, doi:10.3390/w11040667_

Round 1

Reviewer 1 Report

Generally, I review this paper positively but I think it could benefit from several refinements:

- The abstract should contain a preview of the results and more description of the methods.

- The references are not in the correct order as they appear in the manuscript (for example, reference 34 appears in the first paragraph after reference 2, reference 34 should be changed to reference 3 and so on). It could be that the reference 34 actually refers to both reference 3 and reference 4 which in that case, just a typo needs to be addressed.

- I suggest several of the figures would benefit from larger legend sizes and more clear text. Several figures look blurry and are quite small text.

- Additional discussion of these results would be helpful. I suggest the authors create a discussion section which actually unpacks the results in a way that would make them interpretable by stakeholders.

- Where are the calibration statistics that articulate how well the models fit the historical data so that the reader can understand the model performance for extrapolated data?

- You need to also include the fit statistics  for all conditions (linear, poly, and exponential) to ensure the reader understands how you came to your best fit and what that best fit actually means.

- How was the representative solution from each group selected for the MOGA solutions (section 4.2)? Why did you select the representative solution for 1, 2 and 3 the same way but then selected it differently for group 4?

- It’s really not clear how these results could be used by stakeholders. An additional discussion section should be added to digest and interpret the results and also explain the limitations of this work.

Author Response

We thank the reviewer for the time spent on reviewing our paper and suggest valuable comments and thoughtful suggestions. We have tried to address the reviewer’s comments and revised the manuscript accordingly. Our responses are given in the file below and we have highlighted the revised text in red font in the revised manuscript.

Reviewer 2 Report

This paper deals with one interesting topic of these years, that is the nexus between Water, food and energy. The authors present an extension (optimization module WEFSiM-opt) of an already existing simulation model developed by authors themselves useful to assist stakeholders in making informed decisions.

This topic appears original and interesting and seems relevant for readers of the WATER. The goal of the study is well justified in the text. The title clearly describes the article and the abstract reflects its content. Nevertheless, the paper suffers from some weaknesses and in my opinion it’s publishable only after revision.

1) In the Introduction authors just cite the others models of nexus but, to make the work more comprehensible for all readers of the journal, they should write a short description of each models cited and above all write some issue about the functioning and the limits of these models. Without this explanation, for the reader not expert in this area could be difficult understand what authors are dealing with.

2) In my opinion in a work dealing with nationwide simulation model that aim to determine management decisions should be take into consideration also the economic aspect of the components included in the nexus.

For instance, it would be interesting include considerations about the country's wealth, or the PIL per capita, current and expected. It is well know that demand for food (but also for water or energy) is affected also by economic variables.

I’m aware that at this stage of the work is impossible insert in the model economic issues but I suggest to report this suggestion in the introduction and/or at the end of the paper within of the limitations of the work and the future research. To this end I suggest to read one of the few paper that consider also economic aspects of this nexus:

- Peri, M., Vandone, D., Baldi, L. (2017) Volatility Spillover between Water, Energy and Food, Sustainability, 1071.

3) Paraghraph 4.1: “Scenario Development”: Is not very clear which is the source of data used, please indicate in the text or near the figures. Should be interesting integrate date and forecast about the future trend of food demand and supply food demand by Fao Outlook.

Author Response

We thank the reviewer for the time spent on reviewing our paper and suggest valuable comments and thoughtful suggestions. We have tried to address the reviewer’s comments and revised the manuscript accordingly. Our responses are given in the file below and we have highlighted the revised text in the revised manuscript using red font.

Reviewer 3 Report

The manuscript introduced an optimization module to a pre-developed Water-Energy-Food Nexus Simulation Model (WEFSiM). The WEFSiM evaluates the interconnections between the water, energy, and food sectors and provides simulation results in terms of future resource availability and consumption at national scale. Introduction of the optimization module to WEFSiM (WEFSiM-opt) may allow stakeholders to optimize decisions for sustainable resource management. A Single Objective Genetic Algorithm (SOGA) and a Multi-Objective Genetic Algorithm (MOGA) have been investigated for the optimization. This contribution of the paper is interesting, which has been presented with clear methodology and results. However, authors should address the following comments.

1.      Literature review should be more comprehensive, currently there are only 17 references. In particular, the effects of hydro-climatic conditions (e.g. drought) and hydraulic infrastructures (e.g. dams, canals) on the water-energy-food security were not discussed. More reviews (e.g. Craig et al. 2018; Chowdhury et al. 2018; Dang et al., 2018) on these issues should be added.

2.      The decision variables are based on the reliability indices presented in equations (7) and (8). However, authors did not justify the use of these indices instead of other available indices.

3.      Figure 1 should be improved.

4.      Conclusions can be improved.

References:

Craig, MT, Cohen, S., Macknick, J., Draxl, C., Guerra, OJ, Sengupta, M., et al. (2018). A review of the potential impacts of climate change on bulk power system planning and operations in the United States, Renewable and Sustainable Energy Reviews, 98, 255-267.

Chowdhury, A., Dang, TD, Galelli, S., (2018).  Coupling hydrologic and network constrained unit commitment models to understand the water-energy nexus in Laos. American Geophysical Union Fall Meeting, December 2018, Washington D.C., USA.

Dang, TD, Chowdhury, A., Galelli, S., (2018).  Assessing hydrological regime alterations caused by climate change in reservoir-regulated river basins: a case study in the Mekong Basin. American Geophysical Union Fall Meeting, December 2018, Washington D.C., USA.

Author Response

(The authors gave the same response as above.)
